# EVIDENCE-FREE CLAIM VERIFICATION VIA LARGE LANGUAGE MODELS

## ABSTRACT

Hallucination detection is essential for reliable LLMs. Most existing fact-checking systems retrieve external knowledge to verify hallucinations. While effective, these methods are computationally heavy, sensitive to retriever quality, and reveal little about an LLM inherent fact-checking ability. We propose an evidence-free claim verification task: identifying factual inaccuracies without external retrieval. To study this setting, we introduce a comprehensive evaluation framework covering 9 datasets and 18 methods, testing robustness to long-tail knowledge, claim source variation, multilinguality, and long-form generation. Our experiments show that traditional uncertainty quantification methods often lag behind detectors based on internal model representations. Building on this, we develop a probe-based approach that achieves state-of-the-art results. To sum up, our setting establishes a new path for hallucination research: enabling lightweight, scalable, and model-intrinsic detection that can facilitate broader fact-checking, provide reward signals for training, and be integrated into the generation process.

## 1 INTRODUCTION

One of the central limitations of Large Language Models (LLMs) is their tendency to hallucinate — generate facts that are factually incorrect (Huang et al., 2025; Maynez et al., 2020). Such errors are persistent and systematic: theoretical analyses and empirical studies demonstrate that hallucinations are rooted in the fundamental limits of statistical learning and generative modeling (Xu et al., 2024). Moreover, they pose significant social risks, undermining the trust and restricting the safe deployment of LLMs in high-stakes domains such as medicine, law, and science (Asgari et al., 2025). These risks make the development of reliable hallucination detection methods a key research priority in the foreseeable future (Farquhar et al., 2024).

Recent progress in fact-checking has been driven primarily by retrieval-based pipelines, which first retrieve evidence from a database and then verify the correctness of a claim against it, as in FActScore (Min et al., 2023) or SAFE (Wei et al., 2024b). While effective, such methods introduce several challenges: (i) retrieval-based approaches increase latency, as each generation requires querying external databases; (ii) the quality of retrieval is crucial-noisy or irrelevant results can undermine the entire RAG pipeline and lead to missed or false detections (Cuconasu et al., 2024); (iii) retrieval-augmented generation (RAG) inherently prioritizes information from the retrieved context, rather than leveraging the full breadth of the LLM parametric knowledge. As a result, the retrieval-based hallucination detectors are sensitive both to retrieval errors and the model balancing of external versus internal knowledge, limiting their scalability and robustness in practice.

By contrast, LLMs already encode substantial factual knowledge in their parameters through large-scale pretraining on massive and diverse corpora and fine-tuning on targeted datasets. Several studies demonstrate that LLMs store and retrieve encyclopedic and commonsense facts with remarkable accuracy, allowing them to generate factually correct statements even without explicit external grounding (Wang et al., 2020; Kadavath et al., 2022). This intrinsic competence suggests that hallucination detection need not always rely on external evidence: LLMs on their own are capable of identifying factual errors.

Therefore, in this work, we propose an alternative setting: **evidence-free claim verification**. In this formulation, the task is to identify whether the claim is true or false directly — without access to external retrieval. To investigate this setting, we conduct a systematic comparison of 18 methods

Figure 1: The task setting of evidence-free claim verification. Claims from any source (human or LLMs) can be verified without having access to a knowledge base.

across 9 datasets, evaluating different aspects of generalization. We further introduce **INTRA**, a probing approach, which achieves SoTA results and demonstrates strong robustness across datasets.

This shift positions hallucination detection as a lightweight, model-intrinsic capability that exposes what an LLM can reveal about the factuality of its own outputs. We cast the proposed methods as a *factuality-oriented reward models*, akin to those used in other domains to guide and evaluate behavior (Stiennon et al., 2020; Christiano et al., 2017). In this role, the detector can be integrated directly into the generation process to enhance factuality in downstream tasks, improving reliability without relying on external retrieval. We release code and models to support future research.[1]

The contributions of the paper are as follows:

1. We formalize the evidence-free claim verification task, where truthfulness is identified without external retrieval.

2. We establish a large-scale evaluation schema spanning 9 datasets and 18 methods, designed to stress-test robustness across long-tail knowledge, claim source variation, multilinguality, and long-form generation, and use it to provide a systematic analysis that highlights the strengths of internal-based and verbalized approaches.

3. We propose an internal-based claim verifier **INTRA** that achieves SoTA performance across benchmarks and shows strong generalization and robustness.

## 2 RELATED WORK

Traditional approaches to hallucination detection rely mainly on RAG systems. These systems verify the output of models by checking them against external knowledge sources (Min et al., 2023; Wei et al., 2024a; Aushev et al., 2025). FactScore (Min et al., 2023) breaks down the generated text into atomic facts and then calculates how many of them are actually supported by reliable sources such as Wikipedia. While this approach works quite well, RAG-based methods have several drawbacks: they require significant computational resources, their performance strongly depends on retrieval quality, and they are limited by the coverage of external knowledge bases. These limitations prevent them from fully utilizing the model's own parametric knowledge.

To address these limitations, another promising direction has emerged that uses LLM internal representations without needing any external retrieval. SAPLMA (Azaria & Mitchell, 2023) showed that simple linear classifiers trained on hidden layer activations can distinguish true statements from false ones with 60-80% accuracy. Orgad et al. (2025) found that information about truthfulness tends to concentrate in specific tokens and layers. Interestingly, models might encode correct answers internally even when they generate incorrect ones. However, these methods often struggle when applied to different domains than they were trained on.

A related line of research focuses on uncertainty quantification methods, which try to analyze how confident a model is. Traditional approaches often mix different types of uncertainty that are not really related to factuality (Farquhar et al., 2024). Claim Conditioned Probability (CCP) (Fadeeva et al., 2024) addresses this by separating uncertainty about claim values from uncertainty about surface forms and shows particularly good performance across different languages. Among recent

---

[1]https://anonymous.4open.science/r/HalluDetect-2D44

supervised methods, UHead (Shelmanov et al., 2025) uses trainable attention-based heads, while other approaches rely on token-level Mahalanobis distance (Vazhentsev et al., 2025b).

RAUQ (Vazhentsev et al., 2024) takes a different approach by identifying attention heads that consistently drop their activation when the model generates incorrect information. Additionally, several contrastive and self-correction approaches have been proposed. Contrastive methods such as CCS (Burns et al., 2023) use contrastive learning objectives to learn representations of truthfulness, while DoLa (Chuang et al., 2024b) improves truthful generation by contrasting different model layers during inference. Self-correction approaches try to iteratively refine outputs, though detailed analysis by Kamoi et al. (2024) shows that self-correction without external feedback typically does not work well. In this context, our factuality detector can be understood as a specialized type of reward model (Stiennon et al., 2020; Christiano et al., 2017) that focuses on evaluating truthfulness.

## 3 APPROACH

### 3.1 TASK DESCRIPTION

We define the task of **evidence-free claim verification** below. Let a claim be a declarative statement represented by a sequence of tokens $\mathbf{y} = y_1, \ldots, y_n$. The objective is to produce a truthfulness score $s \in [0, 1]$ that estimates the probability of the claim being factually correct, i.e., $s \approx P(\text{Verified} \mid \mathbf{y})$.

The verification function $f$ must operate without access to *any* external knowledge, including web search results, retrieved documents from a vector database, or any other form of external evidence. The assessment of truthfulness must be based solely on the parametric knowledge encoded within the model $M$ and the internal representations it generates when processing the claim $\mathbf{y}$.

Consider the following claim: *"The Eiffel Tower is located in Paris."* An evidence-free verifier must evaluate this statement without querying an external database for the Eiffel Tower's location. Instead, it must infer the claim's veracity by analyzing the model's internal signals, such as hidden state activations, attention patterns, or output probabilities that arise when processing the text. The output would be a single score (e.g., $s = 0.98$) indicating a high likelihood of the claim being true.

### 3.2 EXISTING METHODS

Although initially motivated by detecting errors in LLM outputs, hallucination detection methods are also applicable to verifying claims from any source. Existing approaches to hallucination detection — and by extension claim verification — can be broadly categorized into *supervised* and *unsupervised* approaches, which leverage either output probabilities or internal signals from the model. Below, we provide an overview of representative baselines from each category.

**Unsupervised methods.** Uncertainty quantification is a widely used signal for hallucination detection methods, based on the assumption that LLMs are less confident when producing incorrect information. In this work, we focus on probability– and internal–based methods (Vashurin et al., 2025). Sampling-based approaches are unsuitable for claim verification without access to both the input prompt (that produced the claim) and the full generation.

We consider several straightforward uncertainty measures: **Sequence Probability (SP)**, which computes the probability of a generated sequence; **Perplexity (PPL)**, estimating the average inverse log-likelihood of tokens; and **Mean Token Entropy (MTE)**, which averages the predictive entropy across tokens (Fomicheva et al., 2020). Among state-of-the-art unsupervised measures, we include **Focus** (Zhang et al., 2023), which propagates uncertainty from previous tokens via attention weights; **Claim-Conditioned Probability (CCP)** (Fadeeva et al., 2024), which conditions on the type and meaning of the claim to ignore surface-form and "what to talk about" uncertainty; **Recurrent Attention-based Uncertainty Quantification (RAUQ)** (Vazhentsev et al., 2025a), which identifies uncertainty-aware attention heads and combines their signals with perplexity; and **Attention Score** (Sriramanan et al., 2024), computing the sum of eigenvalues of attention matrices.

**Supervised methods.** A further category of methods involves training lightweight classifiers on LLM representations as predictors of factuality. Several representative approaches fall into this category. **SAPLMA** trains a linear probe on hidden states collected from the 16th layer model, which was identified as the most informative (Azaria & Mitchell, 2023). **Contrast-Consistent Search**

**(CCS)** employs contrastive training with a margin loss over last-layer embeddings, where negatives are hallucinations and positives are true statements. We adapt this method using a relaxed loss condition (Burns et al., 2023). **Mass Mean Probe (MM)** is a linear probing technique that learns a projection defined by the difference of class means in hidden state space (Marks & Tegmark, 2023). **MIND** improves upon previous probing methods by optimizing both the selection of embeddings and the training configuration of the linear model (Su et al., 2024). **Sheeps**[2] is a probing-based approach that trains lightweight classifiers on hidden states using attention pooling to detect hallucinations in grounded generation tasks (CH-Wang et al., 2024). **Supervised Average Token Relative Mahalanobis Distance (SATRMD)** adapts Mahalanobis distance by computing token-level distances across layers and averaging them over tokens to serve as features for a model (Vazhentsev et al., 2025b). **Information Contribution to Residual (ICR)** tracks changes in hidden states across layers by quantifying each module's contribution to the residual stream (Zhang et al., 2025).

Finally, several methods exploit attention weights. **Trainable Attention-based Dependency (TAD)** models conditional dependencies between generation steps, using attention features to estimate the gap between conditional and unconditional confidence, and propagates uncertainty from earlier tokens to detect long-sequence hallucinations efficiently (Vazhentsev et al., 2024). **UHead** combines multiple unsupervised uncertainty estimators with a trainable Transformer-based head on top of the language model and is specifically designed for hallucination detection in long-form generations (Shelmanov et al., 2025).

### 3.3 INTRINSIC TRUTHFULNESS ASSESSMENT

Although these methods are effective in specific settings, they face several limitations. First, the performance of supervised methods often degrades in out-of-distribution scenarios (Vazhentsev et al., 2025b), which are crucial for real-world applications. Second, methods based on hidden states tend to focus on particular algorithmic features (Zhang et al., 2023; Vazhentsev et al., 2025b), layers (Azaria & Mitchell, 2023; Su et al., 2024), or tokens (Azaria & Mitchell, 2023), which further restricts their generalizability. Third, some methods that rely on attention weights (Vazhentsev et al., 2024; Chuang et al., 2024a; Shelmanov et al., 2025) typically require access to both the input prompt and the full generation, which limits their applicability in scenarios such as claim verification, where no prior knowledge of the input is available. To address these limitations, we propose the **In**trinsic **Tr**uthfulness **A**ssessment (**INTRA**) method, which integrates the most effective insights from prior approaches into a unified and generalizable fact-checking framework.

**Token and layer selection.** Early attempts at supervised hallucination detection relied either on sequence-level embeddings – typically formed by averaging token-level hidden states (Su et al., 2024) – or on the hidden states of the first or last generated token (Azaria & Mitchell, 2023). More recent work has shown that this assumption does not always hold. Instead, it proposes leveraging all token-level hidden states, aggregated using token-level uncertainty scores (Vazhentsev et al., 2025b) or supervised attention pooling (CH-Wang et al., 2024). In our method, we focus on the strong generalization and relying on fitted token-level uncertainty scores may not be the optimal solution.

We compute a sequence-level embedding using a learnable parameter vector $\boldsymbol{\theta}$ for a given sequence $\mathbf{y} = y_1, y_2 \ldots y_N$ of length $N$, with corresponding hidden states $\mathbf{h}_l(y_i)$ for the $i-$th token from the $l-$th layer, following the approach of CH-Wang et al. (2024).

$$\mathbf{h}_l(\mathbf{y}) = \sum_{i=1}^{N} \alpha_{l,i} \mathbf{h}_l(y_i), \quad \alpha_{l,i} = \frac{\exp\left(\boldsymbol{\theta}^\top \mathbf{h}_l(y_i)\right)}{\sum_{k=1}^{N} \exp\left(\boldsymbol{\theta}^\top \mathbf{h}_l(y_k)\right)}. \tag{1}$$

Here, $\alpha_{l,i}$ represents the attention weight assigned to the hidden state of token $y_i$, normalized across the sequence via a softmax.

**Layer-wise truthfulness score.** To perform layer-wise claim verification, we apply a linear classifier with learnable weights $\mathbf{W}$ on top of the sequence-level embeddings from each layer:

$$p_l(\text{Verified} \mid \mathbf{y}) = \sigma\left(\mathbf{W}^\top \mathbf{h}_l(\mathbf{y})\right), \tag{2}$$

---

[2]Please note that "Sheeps" is our own convenient adaptation of the method's name, as CH-Wang et al. (2024) do not assign an explicit name to their approach in the original paper.

where $\sigma(\cdot)$ is the sigmoid function, and $p_l(\text{Verified} \mid \mathbf{y})$ represents the probability that the sequence $\mathbf{y}$ is truthful according to layer $l$. We avoid complicating the training procedure or model architecture to ensure that the layer-wise scores do not overfit to specific patterns and remain broadly generalizable. The layer-wise models are trained using the standard cross-entropy loss.

**Aggregated truthfulness score.** Claim verification probabilities can be computed at various layers within a model. Azaria & Mitchell (2023); Servedio et al. (2025) show that the optimal layer for this task can differ depending on the specific generation task. To effectively integrate information across layers, we follow the approach of Vazhentsev et al. (2025b) and train a regression model on top of the layer-wise probabilities. However, acknowledging that previous work has shown the first and last layers to be less effective, we use only the middle layers. We further argue that raw probabilities are not standardized across layers, which could degrade the performance of the regressor. Therefore, we apply quantile normalization (Amaratunga & Cabrera, 2001) as $q(\cdot)$ to the probabilities before using them in the $\mathcal{L}_2$ regression:

$$\textbf{INTRA}(\mathbf{y}) = \sum_{l \in \mathcal{L}} \beta_l \cdot q\left(p_l(\text{Verified} \mid \mathbf{y})\right) + b, \tag{3}$$

where $\beta_l$ and $b$ are the learnable weights and bias term, respectively, of the $\mathcal{L}_2$ regression model. We split the entire training dataset into two parts: the first is used to fit the parameters $\boldsymbol{\theta}$ and $\mathbf{W}$, while the second is used to fit $\beta_l, l \in \mathcal{L}$ and $b$. We use the layers from the first third to the second third of the model (e.g., layers 11 to 22 for LLaMA 3.1-8B-Instruct). We present an ablation study with the various ranges of layers in $\mathcal{L}$ in Table 2.

## 4 EXPERIMENTAL SETUP

**Datasets.** We study the generalization of hallucination detection across heterogeneous sources and domains. To ensure robustness and provide a comprehensive evaluation, we validate nine diverse datasets that collectively test different dimensions of hallucination detection.

A critical aspect of our evaluation is understanding models' hallucinations across the spectrum of knowledge popularity. **PopQA** (Mallen et al., 2023) provides popularity annotations for each entity. **Wild Hallucinations (WH)** (Zhao et al., 2024) complements this with queries about long-tail knowledge. For both datasets we construct atomic claims and their labeling, detailed procedure is described in Appendix A.

Our evaluation also examines generalization across different generation sources and languages. **X-Fact** (Gupta & Srikumar, 2021) provides multilingual claims across 25 languages. **UHead** (Shelmanov et al., 2025) tests robustness by using claims from long generation made by `Mistral 7b` model. **Common Claims (CC)** (Casper et al., 2023) provides `GPT-3-davinci-002` generated claims with human-annotated labels.

Several datasets leverage structured patterns to provide controlled evaluation scenarios. **Cities** (Marks & Tegmark, 2023) uses template-based statements about city locations. **CounterFact (CF)** (Meng et al., 2022) similarly provides factual recall statements. **Companies (CMP)** (Azaria & Mitchell, 2023) covers diverse factual aspects of organizations including headquarters, activities, and representatives. **AVeriTeC** (Schlichtkrull et al., 2023) is a dataset of real-world claims covering fact-checks by 50 different organizations.

We further apply a filtering procedure to ensure that each claim is both high-quality and self-contained, i.e., it includes all necessary context to be fact-checked without relying on surrounding text. Details about datasets and filtering are provided in Appendix C

**Baselines.** We compare the proposed **INTRA** method against both the supervised and unsupervised approaches earlier described in Section 3.2.

**Models.** We evaluate hallucination detection methods on small open-source models, as they provide white-box access required for internal approaches, they are cheaper to run, promote privacy, and democratize research. Moreover, frontier GPT-like closed-source systems increasingly integrate retrieval into their generation process, making them incompatible with our evidence-free setting.

We use `LLaMA 3.1-8B-Instruct` as the base model for both probability-based and embedding-based methods. We train all methods on the split of PopQA, which is described in

Table 1: Performance of hallucination detection methods in the proposed evidence-free setting, measured by ROC-AUC↑ across nine datasets. **Bold** values indicate the best-performing method for each dataset, the second best is underlined. *Avg* column reports the average score across all datasets, summarizing overall robustness. Datasets represent distinct challenges: PopQA (long-tail knowledge), AVeriTeC (human-made claims), UHead (cross-model long-form generation), Cities/CC/CMP/CF (rule-generated claims), WH (long-form generation with long-tail entities), and X-Fact (multilingual claims).

| **Method** | *long-tail* | *human-made* | *cross-model* | *rule-generated* | | | | *long-form* | *multilingual* | **Avg** |
|---|---|---|---|---|---|---|---|---|---|---|
| | **PopQA** | **AVeriTeC** | **UHead** | **Cities** | **CC** | **CMP** | **CF** | **WH** | **X-Fact** | |
| *Unsupervised Methods* | | | | | | | | | | |
| SP | 78.2 | 67.4 | 69.9 | 95.8 | 65.0 | 74.9 | 70.1 | 70.0 | 53.5 | 71.6 |
| PPL | 73.3 | 57.4 | 62.4 | 94.2 | 72.5 | 75.0 | 65.9 | 59.3 | 57.6 | 68.6 |
| MTE | 65.9 | 66.6 | 60.7 | 56.8 | 69.0 | 64.3 | 52.5 | 66.1 | **61.1** | 62.6 |
| Att. Score | 49.7 | 59.4 | 58.6 | 41.5 | 42.8 | 47.8 | 49.1 | 58.6 | 42.7 | 50.0 |
| RAUQ | 65.6 | 67.3 | 61.2 | 57.0 | 68.4 | 63.6 | 52.4 | 67.3 | 60.9 | 62.6 |
| CCP | 72.5 | 65.1 | 69.1 | 89.6 | 65.9 | 79.3 | 66.5 | 69.1 | 51.3 | 69.8 |
| Focus | 67.3 | 59.4 | 58.0 | 74.2 | 63.1 | 66.5 | 59.8 | 67.8 | 55.8 | 63.5 |
| Verb | 72.8 | 62.9 | **72.8** | 98.6 | 74.3 | 89.5 | 75.4 | 73.8 | 51.9 | 74.7 |
| *Supervised Methods* | | | | | | | | | | |
| UHead | 65.7 | 52.1 | 71.2 | 61.6 | 68.9 | 66.9 | 54.5 | **74.2** | 52.6 | 63.1 |
| MM | 79.5 | 63.0 | 57.4 | **99.9** | 67.9 | 75.1 | **82.1** | 56.3 | 51.6 | 71.4 |
| CCS | 86.6 | 54.1 | 66.5 | 95.9 | 67.9 | 77.6 | 66.2 | 66.5 | 54.4 | 71.1 |
| ICR | 74.9 | 51.0 | 70.9 | 58.3 | 50.7 | 54.9 | 51.4 | 56.5 | 57.7 | 58.5 |
| TAD | 84.4 | 56.5 | 60.1 | 73.1 | 65.3 | 69.6 | 53.6 | 70.6 | 57.8 | 65.7 |
| SATRMD | 81.3 | **68.3** | 62.9 | 86.6 | 69.9 | 78.8 | 60.6 | 64.3 | 53.6 | 69.6 |
| MIND | 88.7 | 66.8 | 64.5 | 91.3 | 70.0 | 71.2 | 66.2 | 65.9 | 50.5 | 70.6 |
| Sheeps | 88.8 | 63.6 | 64.8 | 98.1 | 73.4 | 84.5 | 72.7 | 72.1 | 57.0 | 75.0 |
| SAPLMA | 88.6 | 62.9 | 63.5 | 81.8 | 70.9 | 80.6 | 60.6 | 67.6 | 55.0 | 70.2 |
| INTRA | **89.3** | 66.7 | 66.4 | 99.0 | **75.0** | **93.0** | 79.1 | 73.5 | 57.1 | **77.7** |

detail in Appendix A. For the UHead baseline, we use a model trained following the exact procedures described in the original paper (Shelmanov et al., 2025). Embeddings extraction process and other technical details are described in Appendix B.

**Metrics.** We use ROC-AUC and PR-AUC as our primary metrics to evaluate all detection methods. Both are *threshold-insensitive* measures: ROC-AUC evaluates the ability to rank regardless of the decision threshold, while PR-AUC is more sensitive to detecting hallucinations and less strict on false negatives.

## 5 RESULTS

Table 1 reports the ROC-AUC scores for hallucination detection in the proposed evidence-free setting in nine datasets. The results using the PR-AUC metric are presented in Table 6 in Appendix D, where we also include the Verbalized GPT-4.1 scores to provide approximate upper bounds for some datasets. In our experiments, PopQA is treated as the in-domain dataset, since the supervised methods are trained on its training split, while all other datasets are considered out-of-domain. The results highlight clear differences in robustness between the families of methods.

**In-domain performance.** As expected, most supervised methods significantly outperform unsupervised ones. The only exception is UHead, which is pre-trained but not fine-tuned on PopQA. The highest ROC-AUC is achieved by the proposed INTRA method, which outperforms the second-best Sheeps method by 0.5%.

**Unsupervised methods.** We also found that uncertainty-based approaches, with the exception of SP, generally underperform compared to other methods. The model's raw confidence is not always well aligned with hallucinations on arbitrary inputs, a trend consistent with observations reported in standard uncertainty quantification studies for LLMs (Vashurin et al., 2025).

Across evaluated baselines, we identify the Verbalized assessment as a promising direction with remarkably strong result, that stands out as the best-performing unsupervised approach, which contrasts with recent findings on uncertainty quantification (Vashurin et al., 2025). However, it is significantly more compute-intensive than all other methods. Unlike the Verbalized approach, which requires generating multiple tokens to assess the target sequence, other methods complete the task with a single forward pass. Moreover, it suffers from a very high refusal rate on non-English inputs, with up to 58% of cases resulting in refusals or similar behavior.

**Generalization performance.** Several baselines achieve strong results on specific datasets but fail to generalize across hallucination types. For example, MM performs well on its original benchmark but fails on long-form generations such as UHead or WH. Similarly, the UHead model leads in WH and its own dataset, yet lags behind in all other settings. This result illustrates the broader problem of low generalization capacity in hallucination detection (Levinstein & Herrmann, 2025).

We also observe strong generalization from contrastive objectives, which yield competitive results in the specific settings and suggest that contrastive training can enhance robustness in this task, consistent with prior work in other domains (Moskvoretskii et al., 2024).

Finally, our proposed detector, **INTRA**, addresses these issues by achieving the highest average performance and demonstrating consistent robustness across datasets. While not always the best-performing on individual benchmarks, it performs reliably in every setting. By combining token-level hidden states across layers, it captures rich internal information, which appear crucial for evidence-free detection, consistent with prior findings (Dombrowski & Corlouer, 2024). Overall, the proposed INTRA method outperforms the second-best Sheeps method by 2.7% in ROC-AUC across datasets.

**Saturation and contamination.** We also find that widely used rule-generated datasets are largely saturated, with trends diverging from those observed on other benchmarks. As these datasets have been repeatedly used in prior work (Burns et al., 2023; Marks & Tegmark, 2023), we suspect contamination and overfitting at the method design stage. This is further supported by the MM method, which performs poorly across all datasets except the one on which it was originally introduced, indicating overfitting to leaked benchmarks.

**Downstream applications.** Main results suggest that training the full LLM, particularly with a mix of uncertainty-based objectives, can be highly effective. This is evidenced by the UHead method, which ranks first on long-form generation – its original target domain. Moreover, its strong performance and generalization on WH, a dataset constructed with a different model and focused on long-tail entities, indicate substantial potential for scaling and broader applicability.

## 6 ANALYSIS

**Long-tail performance.** Figure 3a presents results stratified by entity popularity. **INTRA** dominates the performance, indicating that supervised and internal signals enable robust hallucination detection even for rare entities.

The verbalized detector is strongest for rare entities (0–100 group), but its advantage fades with increasing popularity. In contrast, SP improves as popularity grows, while both SP and PPL fail on the rarest entities, revealing their weakness on long-tail knowledge.

**Language analysis.** Figure 3b shows that no single method dominates across languages. Perplexity (PPL) performs best for Spanish, German, Arabic, and Tamil, probably reflecting stronger pre-training coverage and surface-form modeling. **INTRA** leads in morphologically rich languages such as Turkish, Georgian, Italian, and Polish, where internal representations better capture structural variation. **CCP** is strongest for Romanian, Indonesian, and Serbian, mitigating surface-form and topical uncertainty in medium-resource settings. Overall, these results highlight that typological properties and resource availability shape the effectiveness of evidence-free detection methods.

**Claim position influence.** While metrics assume claim independence, datasets like UHead contain multiple claims per LLM generation. For WH and UHead, we therefore computed ROC-AUC by claim position and averaged across generations, shown in Figure 2. There is evidence that hallucination rates vary with position with prior work shows errors accumulate on later claims (Belém et al., 2025; Spataru et al., 2024; Wang & Sennrich, 2020).

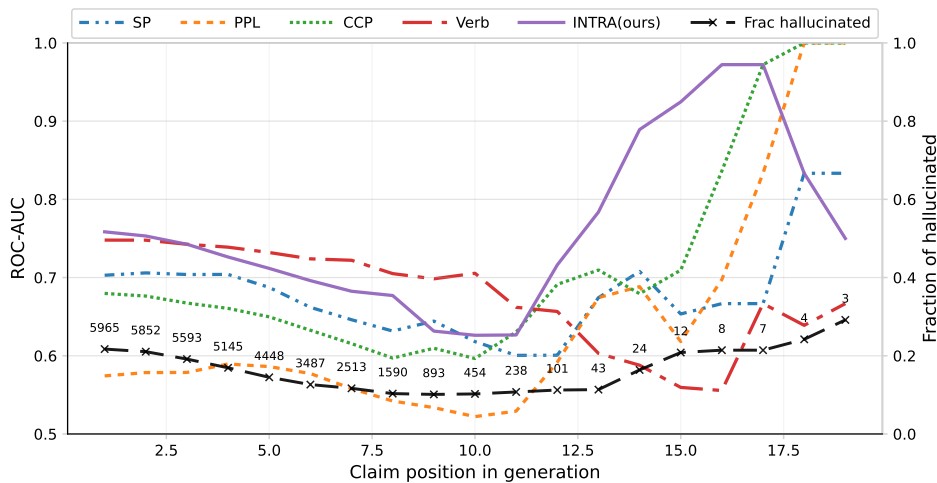

Figure 2: ROC–AUC on WH by claim position. The dashed line shows the fraction of hallucinated claims at each position; the numbers above it indicate the number of claims at that position.

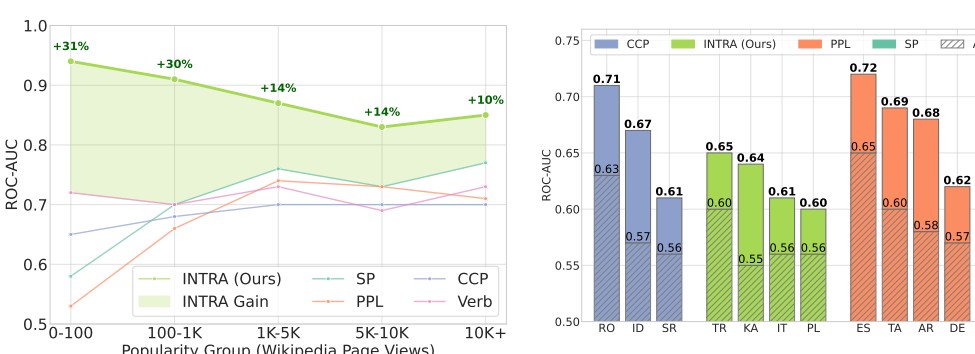

(a) ROC-AUC on PopQA, split into five popularity groups. The green arrow shows the percent improvement of the top method (INTRA) over the runner-up.

(b) ROC-AUC on X-Fact by language. Each column shows the top method, with a hatched overlay indicating the mean of the other methods (SP, PPL, CCP, INTRA).

Figure 3: ROC–AUC splited by popularity and language groups.

Our WH analysis largely confirms this with a U-shaped trend: hallucination rates are high early, drop in the middle, and rise again toward the end. The methods separate well in the early positions, but their differences shrink in the middle. Starting around position ≈12, CCP, Verb, and **INTRA** show a modest recovery, while SP and PPL remain flat. The UHead dataset shows a similar pattern, but with a steeper U-shape, likely due to having fewer claims per generation (see Appendix E).

## 6.1 ABLATION STUDIES

**Performance of individual layers.** Figure 4 presents the ROC-AUC performance of individual layers across various out-of-domain datasets compared to the full INTRA method. The results indicate that, across all tasks, intermediate layers generally yield better performance, consistent with prior work (Azaria & Mitchell, 2023; Vazhentsev et al., 2025b; Servedio et al., 2025). Although the most effective layer varies across datasets, the proposed INTRA method achieves performance equal to or exceeding that of the best individual layer, highlighting the advantage of integrating information across multiple layers.

**Layer selection.** Table 2 presents the results of the INTRA method trained on different subsets of the model's hidden layers. We evaluate using all layers, the first layers (0–8), the last layers (24–32), and various ranges of middle layers. These results demonstrate that training only on the

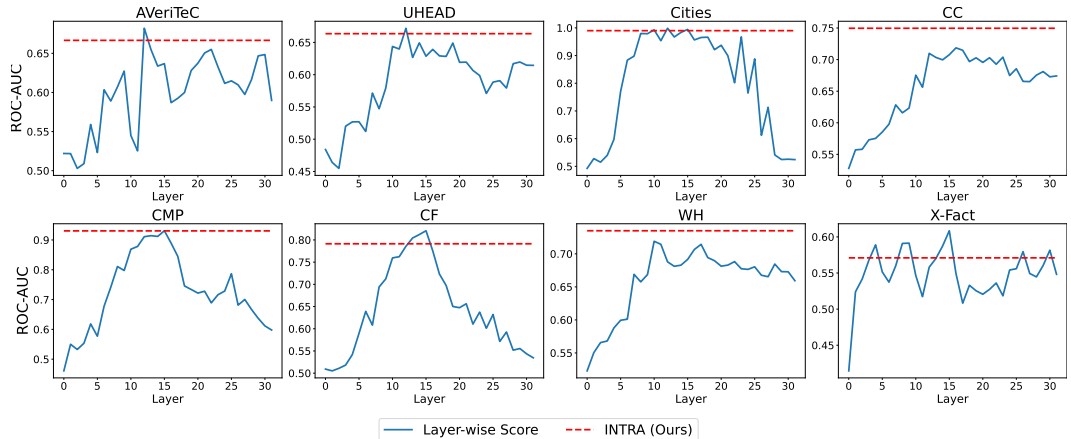

Figure 4: ROC-AUC↑ performance of individual layers in the INTRA method.

Table 2: Impact of different layer subsets on the performance by ROC-AUC↑ of the proposed INTRA method. The best method is in **bold**, the second best is underlined.

| Layers | PopQA | AVeriTeC | UHEAD | Cities | CC | CMP | CF | WH | X-Fact | Avg |
|--------|-------|----------|-------|--------|------|------|------|------|--------|------|
| All | 89.5 | 66.4 | 63.7 | 98.7 | 74.2 | 89.1 | 73.7 | 73.3 | 58.1 | 76.3 |
| 0-8 | 84.9 | 59.5 | 53.0 | 83.9 | 62.1 | 65.4 | 59.7 | 64.1 | 55.2 | 65.3 |
| 24-32 | 86.7 | 63.8 | 61.0 | 73.2 | 71.5 | 70.5 | 58.4 | 70.0 | 57.0 | 68.0 |
| 2-30 | 89.6 | 66.2 | 63.6 | 98.8 | 73.8 | 89.7 | 74.7 | 73.1 | 58.3 | 76.4 |
| 8-24 | **89.6** | **66.8** | 65.3 | **99.3** | 74.4 | 92.0 | 77.4 | 73.5 | 57.6 | 77.3 |
| 11-22 | 89.3 | 66.7 | **66.4** | 99.0 | **75.0** | 93.0 | 79.1 | **73.5** | 57.1 | 77.7 |
| 13-19 | 88.8 | 65.4 | 65.6 | 99.0 | 74.9 | 94.0 | 81.9 | 73.1 | 58.7 | **77.9** |
| 15-17 | 87.5 | 63.0 | 64.8 | 98.7 | 73.8 | **94.1** | **82.8** | 72.2 | **59.8** | 77.4 |
| 16 | 87.2 | 58.7 | 63.9 | 95.7 | 71.9 | 88.9 | 77.6 | 70.7 | 54.9 | 74.4 |

first or last layers is generally ineffective for claim verification tasks. In contrast, training on a small subset of intermediate layers typically yields optimal or near-optimal performance with minimal variation. Importantly, using a single middle layer performs significantly worse than training on a set of middle layers, which additionally underscores the importance of integrating information from multiple layers in a single score.

# 7 CONCLUSION

We introduced the task of **evidence-free claim verification**, systematically comparing 18 methods across nine datasets and proposing the INTRA approach, which achieves state-of-the-art results and strong generalization. Our experiments show that LLMs encode rich factuality signals in their internal representations that can be harnessed without retrieval, enabling lightweight and scalable hallucination detection. Beyond benchmarking, this paradigm opens avenues for integrating truthfulness signals directly into the generation process, serving as reward models for alignment or as monitoring modules in real-world deployments.

Our analyses highlight clear priorities for future work. **First**, *middle layers* emerge as especially informative, suggesting that training should explicitly focus on these representations. **Second**, cross-dataset results reveal *clear winners*: some methods consistently outperform in certain languages and on claims involving rare entities, showing that detector choice matters and that targeted selection can yield substantial gains. **Finally**, incorporating *claim position* as a structural feature of long generations offers another promising direction for improving detection.

ETHICAL STATEMENT

This study relies primarily on open-source language models and publicly available datasets. While we generate a small amount of synthetic data, we apply extensive filtering to ensure quality and to reduce the risk of inappropriate or harmful content. Nevertheless, as with any model-generated data, we cannot guarantee the complete absence of problematic outputs.

Our work focuses on analyzing the signals of models for hallucination detection as a first scientific step. We do not claim that the proposed detectors are fully reliable or suitable for deployment without further validation. The study should therefore be viewed as exploratory research rather than a production-ready solution.

Finally, we emphasize that the proposed detectors are not intended for potential malicious uses. All contributions are released for research purposes only, in accordance with the ICLR Code of Ethics.

REPRODUCIBILITY STATEMENT

We make all efforts to ensure the reproducibility of our results. Nearly all methods in this work are deterministic, with pre-defined hyperparameters specified in the attached code. The full codebase, together with configuration files and scripts to reproduce experiments, is released as supplementary material. All datasets used are either publicly available or provided in processed form, with detailed construction and filtering steps described in Appendix A. With the use of supplementary materials nearly every experiment presented in the paper can be exactly reproduced.

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

## A  DATA CONSTRUCTION

We study the generalization of hallucination detection across heterogeneous sources and domains. To ensure robustness and provide a comprehensive evaluation we propose the use of several datasets as a base for atomic claims:

- **PopQA** (Mallen et al., 2023): This dataset contains simple rule-based questions annotated with a measure of popularity. It was designed to balance popular and less popular entities and to ensure robustness to social and cultural variation. We use it to evaluate model robustness to long-tail knowledge. For this purpose, we split the dataset into training and test sets by stratifying on popularity, so that both sets maintain a similar distribution of popular and less popular entities. To avoid domain adaptation effects, we also ensure that entities and question templates from the training set do not appear in the test set.

    To construct atomic claims, we take the original questions, generate answers with `LLaMA-3.1-8B-Instruct`, and assess correctness using INACCURACY (Moskvoretskii et al., 2025). Each question–answer pair is then converted into an atomic claim with `LLaMA-3.1-70B-Instruct`, the hallucination label is defined by answers InAccuracy.

    The system prompt was "You convert question-answer pairs into factual claims." and the few-shot prompt for a user prompt was as follows:

    > **Converting QA pairs to a Claim Instruction**
    >
    > Convert the question and answer into a factual, declarative English sentence.
    > Examples:
    >
    > Q: What sport does Kwak Hee-ju play?
    > A: Table Tennis → Kwak Hee-ju plays table tennis.
    >
    > Q: What is the religion of James VI and I?
    > A: Protestantism → James VI and I confess protestantism.
    >
    > Q: Who is the father of Sybilla of Normandy?
    > A: Rollo of Normandy → The father of Sybilla of Normandy is Rollo of Normandy.
    >
    > As an answer return just the resulting claim.

- **AVeriTeC** (Schlichtkrull et al., 2023): A human-made fact-checking benchmark with a rigorous verification protocol. We adopt the validation split and use only the claims (without evidence), retaining only refuted or supported instances to reduce bias.

- **UHead** (Shelmanov et al., 2025): An original dataset of questions about popular Wikipedia entities. Atomic claims are extracted with Gemma from long-form generations of a Mistral model, following the CCP setup (Fadeeva et al., 2024). This dataset tests cross-model robustness, since claims are generated by a different model, and also challenges robustness to long-form generation.

- **Cities** (Marks & Tegmark, 2023): a dataset formed of statements from the template "`The city of [city] is in [country]`" using a list of world cities. For each city, authors generated one true statement and one false statement.

- **Companies (CMP)** (Azaria & Mitchell, 2023): A dataset describing various general information about popular companies: headquarters location, what they do, well-known representatives etc.

- **Common Claims (CC)** (Casper et al., 2023): Dataset consists of various statements generated by GPT-3-davinci-002, labeled by humans as being true, false, or neither. The authors left only true/false variables and filtered out some facts to balance the dataset.

- **CounterFact (CF)**: Counterfact was introduced by (Meng et al., 2022) and consists of factual recall statements. The dataset consists of sentences constructed using various patterns: `[X] is [Y] citizen. [X] is owned by [Y]. [X] is the capital of [Y]` etc. Authors adapt Counterfact by using statements which form complete sentences and, for each such statement, using both the true version and a false version given by one of Counterfact's suggested false modifications.

- **Wild Hallucinations (WH)** (Zhao et al., 2024): The original dataset contains 7,917 user queries assuming long-form generation, covering a diverse set of domains. These queries correspond to

Table 3: Examples of filtered data. Claims generated by the model in response to the query: *"Tell me facts about Moby, American musician and songwriter."*

| Claim | Filtered | Verification Explanation |
|---|---|---|
| Moby was born Richard Melville Hall. | No | This fact explicitly mentions the stage name "Moby" and provides a unique personal detail, his birth name. |
| Moby is an American singer-songwriter. | No | This fact explicitly mentions the entity "Moby" and describes his profession, directly tying the fact to him. |
| Moby released his first solo album, *Go*, in 1991. | No | This fact explicitly mentions Moby and provides a specific time frame for the start of his musical career. |
| He rose to international fame. | Yes | The statement is too vague and lacks a specific entity or individual to whom it refers. |
| The album *Play* came out in 1999. | Yes | This fact is vague as it doesn't specify the artist of the album "Play," which could refer to multiple albums with the same title. |
| The album was a commercial success. | Yes | The statement is too vague and could apply to any album, lacking specific identification. |

real information needs but are not found in Wikipedia, thus representing long-tail knowledge. We generate answers to these questions with `LLaMA-3.1-8B-Instruct`, split them into atomic claims, and validate each claim against the provided evidence using FactScore (Min et al., 2023). The resulting atomic claims, labeled for hallucination by FactScore, form a dataset aimed at testing robustness to long-form generation in long-tail knowledge.

• **X-Fact** (Gupta & Srikumar, 2021): The dataset consists of short statements in 25 languages, each labeled for hallucination by expert fact-checkers. It provides a multilingual evaluation benchmark designed to assess both out-of-domain generalization and the capabilities of multilingual models.

## B  TECHNICAL DETAILS

Since our base model was `meta-llama/Llama-3.1-8B-Instruct` an instruction-tuned model, we extracted all claim embeddings using the chat template, with the user prompt set to 'Generate true statement' and the assistant content being the claim text itself.

## C  DATA FILTERING

Due to the lack of context in the task, we had to filter out some claims that could be correct, but did not meet the criteria. Specifically, each claim should describe a unique characteristic of a single entity. As shown in Table 3, some data is accurate, but due to the use of pronouns, it is difficult to verify the accuracy of the information without additional context. Incorrect facts that appear to be correct in form were not filtered out (for example: Moby's album "Go" was released in 1992, not 1991). For filtering out, we used `Llama 3.3 72b Instruct` with prompt C and 10-shot few-shot.

Table 4: Final validation datasets after filtering claims. Cities, Common Claims, CounterFacts and Companies were not filtered out, as the claims are fully consistent with the original task.

| Dataset | Before | After | % filtered |
|---|---|---|---|
| PopQA | 6,974 | 5,494 | 21 |
| AVeriTeC | 424 | 349 | 17 |
| UHead | 2,057 | 921 | 55 |
| Cities | - | 1,496 | 0 |
| Companies (CMP) | - | 1,200 | 0 |
| Common Claims (CC) | - | 4,450 | 0 |
| CounterFact (CF) | - | 1,200 | 0 |
| Wild Hallucinations (WH) | 46,605 | 36,427 | 21 |
| X-Fact | 2,315 | 1,131 | 51 |

---

**Fact Filter Instruction**

You are an expert in verifying atomic facts extracted from Wikipedia. Your task is to determine whether a given fact refers exclusively to the specified topic or entity and cannot reasonably apply to others without direct reference.

**Rules:**

- The fact must be **unique** or **directly tied** to the target entity.
- Generic statements that could apply to many similar entities (e.g., people, places, events) are not acceptable.
- The entity must be **explicitly stated** in the fact.
- Whether the fact is true, false, or debatable is **not relevant**; only specificity matters.
- If the fact clearly names or describes something unique to the topic, it is **GOOD**.
- If the fact is vague, general, or potentially applies to many, it is **BAD**.

**Output format:** Return a brief explanation (1–2 sentences), followed by a semicolon and then either `GOOD` or `BAD`. *Example:* `Mentions a specific architectural style found only in this city; GOOD`

---

To generate a verbalized classification, we used Llama 3.1 8b instruct and a prompt C. Hyperparameters for generation: `temperature=0.3`, `max_tokens=150`, `few_shots=10`. To generate verbalized evaluations, we used *Llama 3.1 8b Instruct* and a prompt with 10 few-shots hyperparameters for generation. Verbalized evaluation may not be stable between generations, so we set a large number of few-shots and a low temperature. Besides, we conducted an additional ablation study on 5-seed generation. As we can see from Table 5, the std between generations is low, which proves the stability of the verbalized approach for claim classification.

Table 5: Results of verbalized classification runs across 5 seeds. Mean scores are reported with standard deviation.

| Dataset | PR-AUC ↑ | ROC-AUC ↑ |
|---|---|---|
| PopQA | $0.8541 \pm 0.0016$ | $0.7276 \pm 0.0033$ |
| AVeriTeC | $0.7585 \pm 0.0129$ | $0.6290 \pm 0.0261$ |
| Cities | $0.9743 \pm 0.0014$ | $0.9861 \pm 0.0006$ |
| Companies (CMP) | $0.6694 \pm 0.0030$ | $0.7431 \pm 0.0033$ |

> **Verbalized Claim Classification**
>
> You are an expert fact-checker with comprehensive knowledge across all domains. Your task is to determine whether claims are factually **TRUE** or **FALSE** based on established facts and evidence.
>
> **Rules:**
>
> - Evaluate the factual accuracy of each claim based on verifiable information.
> - Consider the claim exactly as written.
> - Small errors in dates, numbers, or facts render the claim **FALSE**.
> - If a claim contains multiple parts, **all parts must be true** for the claim to be **TRUE**.
> - If any part is incorrect, misleading, or inaccurate, classify the claim as **FALSE**.
> - Provide a brief explanation (1–2 sentences) of your reasoning.
> - Output format: *explanation;* followed by either `"TRUE"` or `"FALSE"`.

# D ADDITIONAL EXPERIMENTAL RESULTS

Table 6 presents the results using the PR-AUC metric. We also report the Verbalized GPT-4.1 scores to provide approximate upper bounds for some datasets. While GPT-4.1 is able to outperform other methods in several cases, it underperforms in specific settings, such as the multilingual X-Fact dataset due to refusals. Moreover, it requires significantly more computational resources and time, making it less practical for direct application in many tasks. Overall, INTRA demonstrates highly robust and consistent performance in PR-AUC, which aligns with the main findings based on ROC-AUC.

Table 6: PR-AUC↑ Values by Method and Dataset (multiplied by 100). In addition, we have added Verbalized GPT 4.1 score classification to provide upper bounds for some of the datasets. The best method is in **bold**, the second best is underlined.

| Method | long-tail PopQA | human-made AVeriTeC | cross-modal UHead | rule-generated Cities | CC | CMP | CF | long-form WH | multilingual X-Fact | Avg |
|---|---|---|---|---|---|---|---|---|---|---|
| *Unsupervised Methods* | | | | | | | | | | |
| MSP | 89.4 | 79.6 | 40.4 | 94.4 | 64.1 | 71.5 | 66.5 | 32.0 | 64.4 | 66.9 |
| PPL | 88.3 | 74.3 | 31.5 | 94.4 | 71.7 | 77.3 | 62.4 | 21.3 | 69.4 | 65.6 |
| MTE | 83.7 | 78.5 | 30.6 | 55.2 | 66.6 | 61.4 | 51.6 | 29.4 | **73.9** | 59.0 |
| Att. Score | 75.1 | 75.0 | 30.6 | 45.6 | 44.6 | 48.0 | 49.4 | 22.9 | 60.9 | 50.2 |
| RAUQ | 83.6 | 79.1 | 31.0 | 55.2 | 66.2 | 61.3 | 51.6 | 30.6 | 73.7 | 59.1 |
| CCP | 87.7 | 78.8 | 44.8 | 90.5 | 65.7 | 79.0 | 64.7 | 32.4 | 65.0 | 67.6 |
| Focus | 83.5 | 75.3 | 32.1 | 71.0 | 62.1 | 63.9 | 57.0 | 31.7 | 68.8 | 60.6 |
| Verb LLaMA3.1 | 85.4 | 75.8 | 40.5 | 97.4 | 66.9 | 84.3 | 67.2 | 32.0 | 66.0 | 68.4 |
| Verb GPT-4.1 | - | **83.4** | **62.8** | - | - | - | - | - | 71.5 | - |
| *Supervised Methods* | | | | | | | | | | |
| UHead | 84.1 | 70.7 | 62.6 | 58.3 | 65.2 | 65.0 | 53.0 | 36.5 | 64.0 | 60.2 |
| CCS | 95.3 | 59.8 | 30.9 | 96.3 | 62.6 | 64.1 | 67.2 | **67.1** | 41.4 | 64.1 |
| ICR | 89.5 | 71.6 | 47.0 | 56.2 | 50.4 | 45.8 | 51.2 | 19.4 | 58.3 | 54.4 |
| TAD | 94.0 | 75.4 | 31.2 | 72.2 | 63.1 | 66.4 | 52.7 | 37.8 | 68.6 | 62.4 |
| SATRMD | 92.3 | 79.2 | 32.4 | 86.6 | 68.4 | 76.8 | 58.1 | 27.2 | 68.2 | 65.5 |
| MIND | 95.8 | **82.0** | 34.5 | 90.9 | 69.8 | 69.2 | 61.9 | 31.9 | 67.0 | 67.0 |
| Sheeps | 94.9 | 76.8 | 35.1 | 97.8 | 72.6 | 83.9 | 66.9 | 37.2 | 69.0 | 70.5 |
| SAPLMA | 96.0 | 73.9 | 35.9 | 80.9 | 69.2 | 81.8 | 57.4 | 32.7 | 67.5 | 66.1 |
| INTRA | **96.2** | 78.6 | 36.7 | **99.3** | **73.5** | **92.5** | **71.1** | 41.8 | 68.6 | **73.1** |

# E    UHEAD CLAIM POSITION ANALYSIS

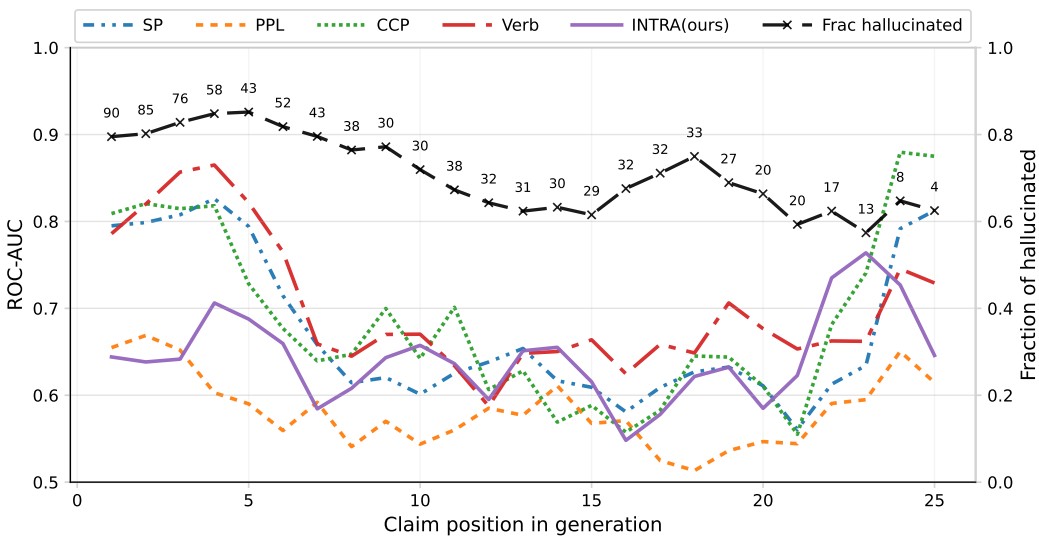

Figure 5: ROC–AUC on Uhead by claim position. The dashed line shows the fraction of hallucinated claims at each position; the numbers above it indicate the number of claims at that position.

## FUTURE DIRECTIONS

Our study establishes a foundation for evidence-free hallucination detection, and several natural directions arise for further work:

- **Extending to more models.** We focus here on `LLaMA 3.1-8B-Instruct` as a representative small, open-source model. Future work can broaden the scope to additional open models of different sizes and architectures, providing a more systematic view of how evidence-free detection scales with model capacity and design choices.

- **Bridging to evidence-dependent systems.** While we emphasize evidence-free methods, it is important to study how they complement RAG and evidence-dependent detectors. Joint evaluations could clarify trade-offs in computational cost, latency, and robustness, and future research might explore hybrid pipelines that combine internal signals with external retrieval.

- **Applications for trust and control.** A promising direction is to integrate evidence-free detectors directly into training or inference pipelines, such as reward signal during alignment, filters during generation, or lightweight modules for user-facing fact-checking. Such applications could help make LLMs not only more accurate but also more transparent and trustworthy in practice.

