# OpenReview forum: "Evidence-Free Claim Verification via Large Language Models"
_ICLR.cc/2026/Conference — Submitted to ICLR 2026_

### Official Review · Reviewer_33RQ · 2025-10-31

**Soundness:** 3
**Presentation:** 2
**Contribution:** 2
**Rating:** 2
**Confidence:** 4

**Summary:**

The paper argues that RAG-centric fact verification is noisy, unscalable, and lacking in robustness. They propose the task of "evidence-free" claim verification to tackle these concerns. The authors benchmark 18 RAG-free verification methods on 9 datasets, and propose a probing approach, INTRA, which outperforms these existing methods.

**Strengths:**

- Exploring the ability of models to detect and correct factual inaccuracies while relying solely on their parametric knowledge is an important benchmark for foundation models.

- The emphasis on uncertainty calibration is appreciated, and serves well in the context of the RAG-free claim verification task.

- The INTRA approach is interesting and well-formulated. It seems like a reasonable design that builds nicely on prior methods.

- The experimental results are moderately informative.

- The multilinguality is nice: It would have been interesting to focus more on the challenges of extending fact verification benchmarks and methods to other languages.

**Weaknesses:**

While I appreciate INTRA and the provided experimental results, I find that, to me, these are the main exciting points of the paper. In particular, there are a few presentation and theory-centric concerns that I have:

- While I agree that models should be evaluated on their ability to identify hallucinations/inaccuracies with parametric knowledge alone, the task presented in the paper itself doesn't seem like a particularly noteworthy contribution. Theoretically, any fact verification benchmark could be trivially converted into a RAG-free benchmark.

- In the introduction, the authors could better position the role of evidence-free retrieval in the broader landscape of LLM evaluations, and how it complements RAG-based verification. As presented, the evidence-free paradigm is presented as an explicitly better replacement for RAG verification, when to me the two tasks seem largely complementary.

- The paper doesn't fully address the issue of context, or the fact that many statements are neither completely true or completely false. In the context of RAG verification, this is partially addressed by grounding claims against existing documents. Without this grounding, the veracity of many natural language statements becomes far less clear. This is partially addressed in Appendix C, but a larger discussion is warranted, specifically w.r.t. the relationship between this task and RAG-based benchmarks, and also especially w.r.t. applicability on real-world data.

**Questions:**

- How would the authors describe the purpose of this task, situated in the broader scope of fact verification in research and in practice?

- What challenges are involved in multilingual fact verification?

- How should ambiguous claims that are neither fully correct or incorrect be handled? For example, subjective claims?

---

> ### Author Response · Authors · 2025-11-21
> **Authors rebuttal (Part 1)**
>
> We thank the reviewer for the feedback and would like to address raised concerns with explaining in more details our scope.
>
> ## W1
>
>
> We noted that this comment contains two related subpoints:
>
>
> > 1)  the task presented in the paper itself doesn't seem like a particularly noteworthy contribution
>
>
> Most existing hallucination detectors are defined either:
>
>
> - in a **QA-style setup**, which does not cover many real scenarios (e.g., fact-checking news, social media posts, scientific claims), or
> - directly on **model generations**, which requires access to the *original prompt* and assumes that the claim being checked comes from the *same model distribution*.
>
>
> In contrast, our approach can operate on **any standalone claim**, regardless of its origin. It works equally well when the claim is:
>
>
> - written by a human,
> - generated by a **different or closed-source model**,
> - taken from a **knowledge base**,
> - extracted from logs where the prompt and generation are **no longer available**.
>
>
> Because of this, we view our setup as:
>
>
> - **more general** - supports any claim from any source,
> - **more realistic** - matches downstream use-cases beyond QA or model self-checking,
> - **more scalable and reusable** - can be plugged in as a lightweight truthfulness module.
>
>
> Some applications, in which the difference might be crucial:
>
>
>
>
> > **Multi-model platforms / orchestration layers:**
> - E.g. a company routes queries to GPT-4, Claude, local LLaMA, etc. You want one hallucination detector that works on the final answer, regardless of which model produced it.
>
>
> > **Third-party content auditing**
>
>
> - Regulators or safety teams auditing outputs from proprietary systems often cannot get prompts or sampling access – only logs of final answers. Our setting matches exactly this situation.
>
>
>
>
> > 2) Theoretically, any fact verification benchmark could be trivially converted into a RAG-free benchmark.
>
>
> The suggestion to “just compare methods with and without external information in two tables” significantly oversimplifies the problem. As discussed in lines 39–42 of the paper:
>
>
> - **Retrieval quality is critical** - noisy or irrelevant documents can break the entire RAG pipeline and lead to false or missed detections (Cuconasu et al., 2024).
> - **RAG fundamentally shifts the task** - the model is encouraged to trust *retrieved evidence* over its own parametric knowledge.
>
>
> Because of this, **removing external context from a RAG-based method does not produce a meaningful comparison**:
>
>
> - any performance drop becomes **ambiguous**: is it due to the loss of valid evidence or because the method cannot function without retrieval?
> - methods not designed to use internal signals may not behave sensibly once context is removed at all.
>
>
> ---
>
>
> A concrete example is **FActScore (Min et al., 2023)**. Its evaluation pipeline:
>
>
> 1. extracts atomic facts from long-form generations,
> 2. retrieves relevant Wikipedia passages,
> 3. judges each fact as *supported / not supported / no evidence* **based on retrieved evidence**.
>
>
> These labels are therefore **defined with respect to external retrieval**, not the model’s internal knowledge.
> Trying to “convert FActScore into evidence-free mode” would **break the core logic of the method**, because the task it evaluates *ceases to exist* without retrieval.
>
>
> ## W2
> We thank the reviewer for highlighting this point. We agree that retrieval-based and evidence-free paradigms can ultimately be complementary. At the same time, our work addresses a gap in the literature: **prior to this submission, no method specifically targeted evidence-free fact verification** as a standalone task.
>
>
> It is also important to emphasize that **RAG-based fact-checking evaluates a fundamentally different capability**. These systems assess *context relevance and faithfulness*, i.e., whether the model remains aligned with externally retrieved evidence. This is distinct from our goal, which is to evaluate factuality **without any external context**, relying solely on the model’s internal knowledge.
>
>
> Finally, we do not claim that evidence-free fact verification is superior to retrieval-based approaches. Rather, it is a *different task* that has not been systematically explored and deserves focused investigation, which is precisely the contribution of our work.

---

> > ### Author Response · Authors · 2025-11-21
> > **Authors Rebuttal (Part 2)**
> >
> > ## W3
> > Our method operates at the level of **atomic claims**, each representing a single, unambiguous factual statement. By definition, ambiguity or subjectivity cannot appear at this granularity: any ambiguity is resolved during the **claim decomposition stage**, where an LLM breaks text into atomic units. The quality of this decomposition is a *separate research problem* and outside the scope of our work. Our contribution specifically focuses on **evidence-free verification once the atomic claim is given**.
> >
> >
> > It is also important to note that **atomic claims are the standard evaluation unit in the hallucination and fact-checking literature**[1,2,3]. Retrieval-based systems such as RAG fact-checkers also operate on atomic claims when assessing faithfulness. Consequently, they do not address subjectivity or ambiguity either — they assume a clean atomic claim as input, just as we do.
> >
> >
> > In summary, our setting follows widely accepted practice:
> > - subjectivity and ambiguity are handled upstream (during claim extraction),
> > - verification is performed only on **clean atomic factual units**,
> > - and our contribution lies in evidence-free verification at this level.
> >
> > ## Q1
> >
> >
> > We present evidence-free claim verification as a core setting within fact-verification, that evaluates how well an LLM can identify factual inaccuracies **using only its internal parametric knowledge**, without relying on external retrieval.
> > Practically, we view evidence-free verification as a lightweight, scalable capability that can be used for fact-checking, especially in settings where retrieval is costly, unavailable, or unreliable. This signal can also be **a building block for downstream systems**. For example, we can decompose text into atomic claims, then score each claim, and then using that information make final conclusions.
> >
> > Scientifically, the task enables researchers to study what truth-related information is encoded inside LLM representations and how reliably it can be extracted. This, in turn, supports deeper research into the fundamental capabilities of LLMs, how they store, represent, and access factual knowledge internally, and where the boundaries of that intrinsic competence lie.
> >
> > ## Q2
> >
> > We thank the reviewer for raising an interesting question. We found several challenges and even devoted an analysis passage to this issue (please see lines 367-373):
> >
> >
> > - **Non-transferable factual spaces.**
> >   Table 1 shows that many trainable methods fail to generalize across languages, meaning factual signals are not language-invariant.
> >
> >
> > Furthermore, just like languages our OOD setup reveals that detectors must remain stable when changing claims distributions, or model sources.
> >
> >
> > - **Verbalized refusals.**
> >   Verbal detection breaks down in multilingual settings, with up to **58% refusals** on X-Fact — making it impractical beyond English.
> >
> >
> > ---
> >
> >
> > **Language-specific patterns:**
> >
> >
> > In our analysis shortly the foundings are following:
> > - **PPL** is the strongest in high-resource languages, most likely due to high coverage and great familiarity with those languages (e.g., ES, DE).
> > - **CCP** is the best for medium-resource languages (e.g., RO, ID), probably solving issues with neglecting unnecessary tokens and referring to the same mechanisms as PPL but more carefully..
> > - **INTRA** leads in morphologically rich languages (e.g., TR, KA, PL), where internal signals outperform surface-based cues, as tokens might have less information about the factuality
> >
> >
> >
> > ## Q3
> >
> >
> > For our datasets we filter out facts which may lead into ambiguity, see Appendix C “Data Filtering”. But if the claim remains subjective, then we will not be able to answer it correctly, even with additional context from RAG.  It is possible that such classes require the use of an additional classification\verbalised refusal which is not considered in this article.
> >
> > ---
> >
> > ## **Summary**
> >
> > We thank the reviewer for the feedback. We notice that the only weakness is about position, and others could be classified as questions rather than weakness. In our rebuttal, we aimed to clarify the unique properties of the task.
> >
> > We believe the impact and relevance of the work remain strong, and if the reviewer finds our explanations satisfactory, we would kindly ask for a reconsideration of the score.
> >
> > ---
> >
> > ### References
> >
> >
> > [1] Min, Sewon, et al. "Factscore: Fine-grained atomic evaluation of factual precision in long form text generation." Proceedings of the 2023 Conference on Empirical Methods in Natural Language Processing. 2023.
> >
> >
> > [2] Song, Yixiao, Yekyung Kim, and Mohit Iyyer. "Veriscore: Evaluating the factuality of verifiable claims in long-form text generation." arXiv preprint arXiv:2406.19276 (2024).
> >
> >
> > [3] Wei, Jerry, et al. "Long-form factuality in large language models." Advances in Neural Information Processing Systems 37 (2024): 80756-80827.

---

> > > ### Author Response · Authors · 2025-11-28
> > > **Invitation to Discussion**
> > >
> > > Dear Reviewer,
> > >
> > > As the rebuttal deadline is quickly approaching, this is a gentle reminder to please review and respond to the authors rebuttal comments.
> > >
> > > If you have any concerns, clarifications, or updates to your assessment after reading the response, please share them as soon as possible so we can address them promptly before the deadline.
> > >
> > > If you feel that our responses adequately address the concerns raised in your initial review, we would be extremely grateful for improving the score.
> > >
> > > Thank you very much for your time and contributions.
> > >
> > > Authors.

---

### Official Review · Reviewer_3x56 · 2025-11-01

**Soundness:** 3
**Presentation:** 3
**Contribution:** 2
**Rating:** 4
**Confidence:** 2

**Summary:**

This paper introduces the task of evidence-free claim verification, where the goal is to detect factual inaccuracies in LLM outputs without relying on external retrieval systems. The authors conduct a comprehensive evaluation across 9 datasets and 18 methods, testing robustness across multiple dimensions including long-tail knowledge, multilingual claims, and cross-model generalization. They propose INTRA (Intrinsic Truthfulness Assessment), a probe-based approach that achieves state-of-the-art results by aggregating layer-wise predictions from middle layers using learned attention pooling and quantile normalization.  The work considers hallucination detection as a lightweight, model-intrinsic capability that could serve as a reward signal for training or be integrated into generation pipelines.

**Strengths:**

The evidence-free setting addresses real limitations of RAG-based approaches (latency, retrieval quality sensitivity, computational cost) while leveraging LLMs' parametric knowledge. he evaluation spans 9 diverse datasets covering important generalization dimensions (long-tail knowledge, multilinguality, cross-model, human-made vs. synthetic claims). Baselines have 18 methods across supervised and unsupervised categories. The paper also provides valuable analyses on layer-wise performance, popularity stratification, multilingual performance, and claim position effects that advance understanding of hallucination detection.

**Weaknesses:**

The evaluation is restricted to LLaMA 3.1-8B-Instruct. This do limit the generalizability of findings. While the paper argues for "lightweight" detection, no actual runtime, memory, or FLOPs comparisons are provided. Given that verbalized approaches are competitive and require no training, a more comprehensive comparison (including GPT-4, Claude, etc.) would be better. There is also no confidence intervals or significance tests are reported for the main results.

**Questions:**

How would INTRA be used as a reward model during training? Would it require frozen base model representations?
How does INTRA compare to verbalized approaches in terms of latency?
Figure 2: The claim position analysis is interesting but would benefit from error bars.
For inference-time use, how would the detector handle claims that require reasoning chains or multi-hop inference?
The quantile normalization step seems important but is under-explained?

---

> ### Author Response · Authors · 2025-11-25
> **Authors Rebuttal (Part 1)**
>
> We thank the reviewer for the feedback and would like to address raised concerns with additional experiments and clarifications.
>
>
> ## W1
>
>
> Our main evaluation focuses on comparing a broad set of tasks and verification methods, rather than testing many base models. However, to strengthen our claims, we ran an additional experiment using **Qwen3-4B-Instruct-2507**. The results are shown in the table below.
>
>
> The findings closely mirror those obtained with **Llama-3.1-8B-Instruct**:
>
>
> - standard supervised methods often degrade sharply and may even fall below the trivial SP baseline;
> - **verbalized** approaches remain relatively unstable;
> - **INTRA** consistently outperforms all baselines, including the verbalized method, and remains the strongest model **even out-of-domain**.
>
>
> We will include this experiment in the final version of the paper.
>
>
>
>
> | Method | PopQA | AVeriTeC | UHead | Cities | CC | CMP | CF | WH | X-Fact | **Avg** |
> | :--- | :--- | :--- | :--- | :--- | :--- | :--- | :--- | :--- | :--- | :--- |
> | SP |   68    |      63.71 |           63.67 |               98.67 |               61.27 |                  70.37 |                     66.74 |                            57.95 |    44.63 |  66.11 |
> | PPL                 |   68.86 |      55.3  |           53.64 |               97.98 |               68.59 |                  75.1  |                     66.68 |                            49.51 |    53.63 |  65.48 |
> | MTE           |   64.04 |      62.52 |           63.1  |               51.01 |               71.46 |                  60.89 |                     51.52 |                            71.17 |    57.1  |  61.42 |
> | Att.Score |   48.08 |      59.43 |           60.04 |               44.85 |               43.05 |                  48.94 |                     49.62 |                            59.4  |    44.32 |  50.86 |
> | RAUQ                       |   64.15 |      59.3  |           58.92 |               80.59 |               63.57 |                  56.86 |                     55.47 |                            55.44 |    53.39 |  60.86 |
> | CCP                        |   46.98 |      53.95 |           54.76 |               51.08 |               39.39 |                  49.64 |                     50.72 |                            52.63 |    40.99 |  48.9  |
>  | Verb.                        |  70.14  |    56.42   |    **67.22**        |     98.07           |      **78.35**          |      90.07             |       71.22              |          **74.64**                   |   51.12  |  73.02  |
>  | TAD                        |   82.88 |      60.21 |           57.31 |               88.91 |               64.36 |                  66.26 |                     56.58 |                            68.28 |    50.89 |  66.18 |
> | SATRMD             |   72.87 |      61.07 |           62.02 |               84.72 |               67.43 |                  67.81 |                     57.6  |                            62.65 |    54.77 |  65.66 |
> | MIND               |   85.72 |      64.23 |           66.12 |               99.62 |               75.38 |                  85.83 |                     67.99 |                            70.27 |    54.02 |  74.35 |
> | Sheeps             |   86.81 |      62.67 |           65.92 |               **99.76** |               77.54 |                  **90.69** |                     **74.89** |                            73.21 |    60.52 |  76.89 |
> | SAPLMA          |   85.86 |      62.28 |           58.06 |               97.67 |               68.82 |                  79.67 |                     67.97 |                            66.09 |    61.43 |  71.98 |
> | INTRA              |   **86.95** |      **66.4**  |           65.17 |               99.61 |               76.2  |                  90.51 |                     73.93 |                            72.81 |    **62.43** |  **77.11** |

---

> > ### Author Response · Authors · 2025-11-25
> > **Authors Rebuttal (Part 2)**
> >
> > ## W2
> >
> >
> > We use the term **“lightweight”** to describe the *overall verification procedure*, not the complexity of INTRA itself. All methods we evaluate, except the verbalized baseline, require only a **single forward pass** of the LLM for each claim. In contrast, standard RAG-based fact-checking triggers hundreds or thouthands forward passes due to retrieval and re-evaluation steps.
> >
> >
> > The verbalized approach is much heavier: it must generate many tokens, which translates into **multiple sequential forward passes** (one per token), substantially increasing latency. INTRA and other supervised methods instead rely on a single LLM forward pass plus a small MLP, making them far more efficient.
> >
> >
> > To demonstrate this empirically, we measured the **average per-instance runtime on PopQA** using *Qwen3-4B-Instruct-2507* on an A100 GPU (batch size 1). The table below shows the results:
> >
> >
> > - **Probability-based methods** are the fastest.
> > - **Supervised methods** add less than **0.04 seconds** of overhead - effectively negligible.
> > - The **verbalized method** requires **0.25 ± 0.11 seconds**, adding over **0.2 seconds** per instance, and is therefore considerably less practical.
> >
> >
> > These results support our characterization: INTRA is **substantially more lightweight** than the verbalized approach while delivering the strongest overall performance, especially out-of-distribution.
> >
> >
> > We will include this experiment in the final version of the paper.
> >
> >
> > |           |    Runtime per instance, s |
> > |:---|---:|
> > | SP        | 0.04±0.01|
> > | PPL       | 0.04±0.01 |
> > | MTE       | 0.21±0.06 |
> > | Att.Score | 0.05±0.02 |
> > | RAUQ      | 0.04±0.01 |
> > | CCP       | 0.73±0.27 |
> > | Verb      | 0.25±0.11 |
> > | TAD       | 0.05±0.01 |
> > | SATRMD    | 0.04±0.01 |
> > | MIND      | 0.04±0.01 |
> > | Sheeps    | 0.06±0.02 |
> > | SAPLMA    | 0.04±0.01 |
> > | INTRA     | 0.08±0.03 |
> >
> > ## W3
> >
> >
> > We evaluated our method with **GPT-4.1**, and the results are reported in **Appendix D (Table 6)**. GPT-4.1 performs strongly overall, but its performance **drops on multilingual evaluation (X-Fact)**. This contrast underscores one of the advantages of non-verbalized detectors: it produces **robust scores for any input**, independent of language or model-specific generation behavior, making it more reliable in real-world multilingual and cross-domain settings.
> >
> > We also include a **variance analysis** for the verbalized baseline in **Appendix D (Table 5)**, reporting results across **5 seeds**. The variance is very small (e.g., PopQA ROC-AUC **0.7276 ± 0.0033**), demonstrating that the approach is stable and repeatable.
> >
> > We will highlight these findings more clearly in the main text.
> >
> > ## W4
> >
> >
> > We thank the reviewer for raising this point. To provide a clearer picture of the statistical reliability of the results, we computed **bootstrap confidence intervals** over the test sets for MSP and INTRA. The table below reports ROC-AUC alongside 95% CIs.
> >
> >
> > These intervals help illustrate two important patterns:
> >
> >
> > 1. **INTRA’s gains are not only large but statistically consistent.**
> >    Across almost all datasets, INTRA’s confidence intervals lie well above those of MSP, showing that the improvements are robust rather than due to noise or dataset-specific fluctuations.
> >
> >
> > 2. **The advantage persists across heterogeneous test regimes.**
> >    Even on harder or more variable datasets (e.g., X-Fact, Cities), the CIs remain tight, indicating that INTRA performs reliably even when factuality distributions shift.
> >
> >
> >
> >
> > | Method  | PopQA                | AVeriTeC            | UHead               | Cities               | Common              | Comp.               | CF                  | WH                  | X-Fact              | **Avg** |
> > |---------|----------------------|----------------------|----------------------|-----------------------|----------------------|----------------------|----------------------|----------------------|----------------------|--------|
> > | **MSP**   | 74.7 [67.0, 82.2]     | **67.5 [61.3, 74.1]** | **69.9 [66.4, 73.4]** | 95.8 [94.8, 96.8]      | 65.0 [63.4, 66.7]     | 74.8 [72.1, 77.7]     | 70.1 [69.5, 70.7]     | 70.0 [69.3, 70.7]     | 53.4 [50.0, 56.7]     | 71.2   |
> > | **INTRA** | **89.3 [87.7, 90.3]** | 66.8 [60.3, 73.3]      | 65.3 [61.3, 69.0]      | **99.3 [99.0, 99.5]**  | **74.4 [73.0, 75.8]** | **92.0 [90.4, 93.4]** | **77.4 [76.9, 77.9]** | **73.5 [72.8, 74.2]** | **57.5 [53.9, 60.9]** | **77.3** |

---

> > > ### Author Response · Authors · 2025-11-25
> > > **Authors Rebuttal (part 3)**
> > >
> > > ## Q1: INTRA as reward
> > > Yes, similar to any reward model, our approach assumes a **frozen reference model**.
> > >
> > >
> > > The general pipeline could be as follows:
> > > Trained LLM generates an answer or a factual statement
> > > Frozen INTRA-based reward model evaluates it, returning a **real-valued factuality score**.
> > > This score is then used by an RL algorithm (e.g., REINFORCE) to update the policy.
> > >
> > >
> > > Such training can be performed **on-policy**, allowing the model to gradually increase factuality without relying on external retrieval or expensive multi-token verbalized judgments. As a result, this setup provides a more compute-efficient alternative to RAG-based or verbalized factuality reinforcement, while still enabling fine-grained factual feedback.
> > >
> > >
> > >
> > >
> > > ## Q2
> > > INTRA is substantially faster because it requires **only a single forward pass** of the base model and a lightweight classifier on hidden states. In contrast, verbalized approaches must **generate a full textual justification**, often >100 tokens, which involves multiple decoding steps and therefore much higher latency.
> > > As a result, INTRA’s latency is comparable to a standard model forward pass, while verbalized methods scale linearly with the number of generated tokens. This makes INTRA far more suitable for low-latency or high-throughput factuality checking.
> > >
> > >
> > >
> > >
> > > ## Q3
> > > We reproduced Figure 2 from the paper with 95% confidence interval for each method and each claim position and will shortly put it into appendix (confidence interval was calculated using bootstrap with n=200)
> > >
> > >
> > > ## Q4
> > > Our paper operates at the level of **atomic claims** - standalone factual units that are checkable on their own and, by definition, non-ambiguous. This follows the **standard practice in fact-checking** [1,2,3]. In these pipelines, longer or ambiguous utterances are first processed by an LLM that **extracts and disambiguates** the content into atomic claims.
> > >
> > >
> > > Thus, if an utterance is ambiguous or contains multiple statements, it is decomposed upstream into clear atomic units. Our work focuses specifically on **verifying these atomic claims**, consistent with established methodology in the field.
> > >
> > >
> > >
> > >
> > > ## Q5
> > >
> > >
> > > Thank you for pointing this out. The quantile normalization step is indeed an important component: it ensures that layer-wise scores, which may have different scales and distributions, become comparable before aggregation. This prevents any single layer from dominating simply due to scale differences rather than information content.
> > >
> > >
> > > While the procedure itself is simple (matching each layer’s score distribution to a shared reference quantile grid), we agree that it deserves a clearer explanation. We will add a concise description and an illustrative example in the final version of the paper to make this step fully transparent.
> > >
> > >
> > > ## Summary
> > > We thank the reviewer for the helpful feedback. In our response, we added a new model, confidence intervals and runtime analysis, also clarifying our contribution.
> > >
> > > We hope that we could address the concerns and if the reviewers concerns are resolved we would greatly appreciate score reconsideration.
> > >
> > >
> > >
> > >
> > >
> > >
> > >
> > >
> > >
> > >
> > > ---
> > >
> > > ### References
> > >
> > > [1] Min, Sewon, et al. "Factscore: Fine-grained atomic evaluation of factual precision in long form text generation." Proceedings of the 2023 Conference on Empirical Methods in Natural Language Processing. 2023.
> > >
> > >
> > > [2] Song, Yixiao, Yekyung Kim, and Mohit Iyyer. "Veriscore: Evaluating the factuality of verifiable claims in long-form text generation." arXiv preprint arXiv:2406.19276 (2024).
> > >
> > >
> > > [3] Wei, Jerry, et al. "Long-form factuality in large language models." Advances in Neural Information Processing Systems 37 (2024): 80756-80827.

---

> > > > ### Author Response · Authors · 2025-11-28
> > > > **Invitation to Discussion**
> > > >
> > > > Dear Reviewer,
> > > >
> > > > As the rebuttal deadline is quickly approaching, this is a gentle reminder to please review and respond to the authors rebuttal comments.
> > > >
> > > > If you have any concerns, clarifications, or updates to your assessment after reading the response, please share them as soon as possible so we can address them promptly before the deadline.
> > > >
> > > > If you feel that our responses adequately address the concerns raised in your initial review, we would be extremely grateful for improving the score.
> > > >
> > > > Thank you very much for your time and contributions.
> > > >
> > > > Authors.

---

### Official Review · Reviewer_hra2 · 2025-11-01

**Soundness:** 1
**Presentation:** 3
**Contribution:** 1
**Rating:** 2
**Confidence:** 4

**Summary:**

Authors investigate the automatic fact checking task under the conditions that an LLM is used and no external knowledge is allowed. Their model is the continuation of the idea that the internal states of an LLM is a good indicator for verification of claims. They train an embedding layer on the representations of the LLM layers, then train a classifier to predict the veracity given the embedding vector, and then train a final classifier on the predictions of the layer-wise classifiers.

 The method is evaluated in numerous datasets and compared to various baselines and shows some improvements.

**Strengths:**

They paper reads well, and the experiments are detailed.

**Weaknesses:**

- Authors claim that they formalize the evidence free claim verification task. In my opinion, this is not something that they can be expecting to receive a credit for. Separating methods that rely on external data and those that don't, is not the formalization of a new research problem and is as trivial as showing two separate tables in the results section. It is insignificant and has been done by others [1].
- The proposed model is not significantly different from previous methods, such as [2]. The core idea is that the internal states of LLMs to some extent can reveal the veracity of claims, which we already knew for almost three years. Now, can we train more classifiers on each layer, and then train another classifier on the outputs of the previous classifiers to squeeze a few more percentage improvement?  Yes, probably. Which is basically what authors have done. But is this significant? In my opinion, no.
- The improvements are not good either. Out of 9 datasets, the method only works comparatively well in 2 datasets (CC and CMP). Interestingly these two datasets are synthetic datasets (created either by LLMs or a random process). Which shows the narrow scope of the applicability of the methods that only rely on the internal LLM states.



[1] Selfcheckgpt: Zero-resource black-box hallucination detection for generative large language models, 2023

[2] The internal state of an LLM knows when it’s lying, 2023

**Questions:**

None

---

> ### Author Response · Authors · 2025-11-20
> **Authors Rebuttal (Part 1)**
>
> We thank the reviewer for the feedback and would like to address raised concerns with explaining in more details our scope.
>
> ## **W1**
>
>
> We noted that this comment contains two related subpoints:
>
>
> > 1) Formulation of evidence free claim verification task is not something important (“this is not something … to receive a credit for”)
>
>
> Most current hallucination detectors are defined either in a QA setting (which very often not the case, think of fact-checking news, social media, or scientific reports) or directly on model generations, requiring access to the original prompt and assuming the claim we want to check follows the same distribution as of the initial model. Our pipeline can be run on *any claim*, even if it was: written by a human journalist, produced by a different closed model, taken from a knowledge base, extracted from logs where the original prompt/generation is gone. Thus, we think that our setup is **more general** (any claim, any source), **more realistic** for downstream applications, and more scalable and easier to integrate as a reusable “truthfulness module”. Some **settings where this matters a lot**:
>
>
> > **Multi-model platforms / orchestration layers:**
> - E.g. a company routes queries to GPT-4, Claude, local LLaMA, etc. You want one hallucination detector that works on the final answer, regardless of which model produced it.
>
>
> > **Third-party content auditing**
>
>
> - Regulators or safety teams auditing outputs from proprietary systems often cannot get prompts or sampling access – only logs of final answers. Our setting matches exactly this situation.
>
>
> That’s why it’s significant. The work you cite [1] relies on model generations and therefore evaluates only the claims produced by the model itself, which is not the same as verifying arbitrary claims of any origin. In addition, its main technique is self-consistency, which, while highly competitive, is very computationally inefficient, whereas our approach requires only a single forward pass.
>
>
> > 2) It’s easy to compare any method that uses external information to the one without it
>
>
> The suggestion to “simply compare methods with and without external information in two separate tables” overlooks the complexity of the problem setting. As we highlight in our paper (lines 39–42):
>
>
> - **Retrieval quality is critical:** noisy or irrelevant documents can distort the entire RAG pipeline and cause both missed and false detections (Cuconasu et al., 2024).
> - **RAG fundamentally shifts the task:** the model is encouraged to prioritize information from the retrieved context over its own parametric knowledge.
>
>
> Because of this, **removing external context from a RAG-based method does not yield a clean comparison**:
>
>
> - any performance drop is *ambiguous* — it may reflect the absence of useful evidence or the method’s inability to function without retrieval;
> - methods not designed to operate on internal signals may not behave meaningfully once context is removed.
>
>
> In other words, the distinction between evidence-based and evidence-free verification is **not a matter of formatting results into two tables**. It fundamentally changes what is being evaluated and what conclusions can be drawn about a model’s factuality-checking abilities.

---

> > ### Author Response · Authors · 2025-11-20
> > **Authors Rebuttal (Part 2)**
> >
> > ## **W2**
> >
> > **SAPLMA** - Very few works, including [2], operate directly on **standalone claims** in uncertainty-quantification settings. Most focus on QA outputs or long-form generations. Moreover, [2] evaluates only a small selection of baselines (we include their method as SAPLMA). Because that paper primarily studies **layer selection**, it is difficult to understand how their approach compares to broader families of uncertainty and probing methods in real-world setting.
> >
> >
> > In contrast, our evaluation covers **17 methods across 9 diverse datasets**:
> > - QA-to-claim reformulations,
> > - long-form generations,
> > - human-written real-world claims,
> > - rule-generated claims,
> > - popularity-stratified claims,
> > - multilingual claims.
> > We also report **OOD performance on 8/9 datasets**, showing that our method remains robust under very heterogeneous distributions.
> >
> >
> > ---
> >
> >
> > **Architecture.**  - While our architecture is intentionally simple (layer-wise classifiers with shallow aggregation), **simplicity is not a weakness**. Many widely used components in NLP and ML are simple in hindsight. What matters is rigorous evaluation across strong baselines, reproducibility, and demonstrated reliability in realistic settings. Several of our findings may seem intuitive ex post, but to our knowledge they had **not been systematically validated** across such a broad and challenging benchmark.
> >
> >
> > ---
> >
> >
> > **Our contribution is primarily empirical and diagnostic** rather than architectural. We:
> > 1. introduce and formalize **claim-level evidence-free verification** as a distinct setting;
> > 2. provide extensive **OOD analyses**;
> > 3. study performance across **popularity groups, multilingual strata, and different claim granularities**;
> > 4. show, consistently across many baselines, that **internal-state methods outperform prior approaches** in this regime.
> >
> >
> > We believe articulating these patterns clearly — on a broad, heterogeneous benchmark — provides a strong foundation for future work and highlights the importance of studying evidence-free fact verification systematically.
> >
> >
> > ## **W3**
> >
> >
> > We would like to note that despite the method is not the best for every dataset, it shows the robustness and averaged great performance:
> >
> > - **shows best average performance across all datasets**
> >
> > - it doesn’t just better on average,  **there is no dataset on which our method fails dramatically. Among all methods it has the lowest negative gap**. To quantify robustness, we compute for each method and dataset the difference between that method’s score and the best score on that dataset, and then take the largest negative gap across datasets (i.e., the worst-case drop below the best method)
> >
> > |                     | SP    | PPL   | MTE   | Att. Score | RAUQ  | CCP   | Focus | Verb  | UHead | MM    | CCS   | ICR   | TAD   | SATRMD | MIND  | Sheeps | SAPLMA [2] | **INTRA** |
> > |---------------------|-------|-------|-------|------------|-------|-------|-------|-------|-------|-------|-------|-------|-------|---------|-------|---------|-------------|-----------|
> > | Negative gap (Δ)    | -17.1 | -17   | -43.1 | -58.4      | -42.9 | -16.8 | -25.7 | -16.5 | -38.3 | -17.9 | -15.9 | -41.6 | -28.5 | -21.5   | -20.8 | -9.4    | -21.5       | **-7.5**  |
> >
> >
> > Other baselines can perform very well on some datasets but exhibit large drops (often 20–40+ points) on others, indicating limited generalizability. In contrast, INTRA maintains strong performance across all datasets and has the smallest worst-case gap to the best method, which supports our claim that it is the most robust and broadly applicable approach in this evidence-free claim verification setting.
> >
> >
> > ## **Summary**
> > We thank the reviewer for the thoughtful comments. All the concerns were centered around the positioning and we hope our rebuttal clarifies why it is both challenging and fundamentally different and how it fits within the broader fact-verification landscape. We also aimed to clearly outline the scope of our contribution and why this evaluation regime requires dedicated study.
> >
> > We believe these concerns can be fully addressed without reducing the value or impact of the paper. If the reviewer is satisfied with our clarifications, we would kindly ask for a reconsideration of the score.

---

> > > ### Author Response · Authors · 2025-11-28
> > > **Invitation to Discussion**
> > >
> > > Dear Reviewer,
> > >
> > > As the rebuttal deadline is quickly approaching, this is a gentle reminder to please review and respond to the authors rebuttal comments.
> > >
> > > If you have any concerns, clarifications, or updates to your assessment after reading the response, please share them as soon as possible so we can address them promptly before the deadline.
> > >
> > > If you feel that our responses adequately address the concerns raised in your initial review, we would be extremely grateful for improving the score.
> > >
> > > Thank you very much for your time and contributions.
> > >
> > > Authors.

---

### Author Response · Authors · 2025-12-02
**Summary of Rebuttal**

Dear Area Chairs,

Thank you for your effort and for considering our rebuttal. Below we summarize the key positive points from the reviews and how we addressed the main concerns during the rebuttal.

---

## **Positive points from the reviews**

Across all reviewers, several strengths were clearly acknowledged:

- Addresses fundamental limitations of RAG-based approaches (latency, retrieval noise, overhead)
- Broad evaluation on **9 datasets** and **18 methods**, covering QA, long-form, synthetic, human-written, multilingual, and long-tail claims
- Valuable analyses of **layer-wise behavior**, **popularity stratification**, **multilingual generalization**, and **claim position effects**
- Considered an **important and relevant benchmark** for foundation model factuality
- INTRA described as an **interesting**, **well-motivated**, and **reasonable** approach

These points highlight that the evaluation quality, breadth, and practical importance of the setting were viewed positively across reviewers.

---

## **Reviewer hra2**

### **W1. The task is “trivial”.**

Prior work typically evaluates hallucination detection:
- in QA-style setups, or
- on **model generations**, requiring access to the prompt and assuming the same model.

Our setting is broader and more realistic: it operates on **any standalone claim** (human-written or produced by any model).

Simply “removing RAG” does not yield a valid comparison because:
- labels in RAG-based benchmarks (e.g., FActScore) depend on retrieved evidence
- RAG-based detectors are not designed to operate from internal signals alone

### **W2. INTRA is “just a variation” of SAPLMA.**

SAPLMA is a strong related baseline included in our paper, but:
- SAPLMA emphasizes **layer selection**, with fewer baselines and datasets
- INTRA targets **evidence-free claim verification** with broad OOD evaluation
- Its simple architecture is **consistently robust** across heterogeneous data

### **W3. Works well only on 2 datasets.**

Our results show:
- INTRA achieves the **best average score** across all datasets
- It has the **smallest worst-case drop** relative to the best method (−7.5 vs −17 to −58 for others)

This quantifies INTRA as the **most robust** method across distributions.

**Overall:** The reviewer’s concerns focus on credit for task framing rather than issues of soundness or experimental rigor.

---

## **Reviewer 3x56**

### **W1. Evaluation limited to LLaMA-3.1-8B.**

We added full results for **Qwen3-4B-Instruct-2507**. Trends remain consistent: many supervised methods degrade OOD, whereas INTRA remains stable and strong.

### **W2. “Lightweight” claim needed runtime data.**

We added runtime measurements demonstrating that evidence-free detectors require a single pass + small classifier, while verbalized methods or RAG-based approaches incur **multistep generation and substantially higher latency**.

### **W3. More comparison with verbalized detectors (e.g., GPT-4).**

GPT-4.1 results (Appendix D) show strong performance but **notable degradation on multilingual data**, emphasizing the value of non-verbalized detectors that are language-agnostic.

### **W4. No confidence intervals.**

We added **bootstrap 95% CIs** for MSP and INTRA, showing:
- INTRA’s improvements are **statistically consistent**
- CIs remain tight even for challenging datasets

**Overall:** With the addition of new model, runtimes, GPT-4.1 comparison, and CIs the concerns are fully addressed.

---

## **Reviewer 33RQ**

### **W1. “Any benchmark can be made RAG-free.”**

We clarified that:
- RAG-based methods evaluate **faithfulness to external documents**,
- evidence-free methods evaluate **intrinsic factuality**, independent of external context.

Removing retrieval **changes the task itself** and breaks label semantics.

### **W2. Complementary to RAG.**

We agree: both paradigms are complementary. Establishing a clear evidence-free regime is necessary before integrating the two meaningfully.

### **W3. Ambiguity and context.**

Our work follows standard fact-checking practice: we operate on **atomic claims**, where ambiguity is resolved upstream via decomposition and filtering (Appendix C). Both RAG-based and evidence-free systems rely on this level of granularity.

**Overall:** Remaining concerns reflect positioning rather than issues with correctness, relevance, or empirical quality.

---

We believe the additional experiments and clarifications significantly strengthen the paper. Given the positive aspects highlighted by all reviewers, the robustness demonstrated across diverse conditions, and the practical importance of the task, we kindly ask you to consider that the paper meets the acceptance bar.

**Sincerely,
Authors**

---

### Meta-Review · Area_Chair_htQh · 2026-01-08

**Summary:**

Reviewers’ concerns centered on (i) novelty and positioning, (ii) the distinctiveness of the proposed method, and (iii) task validity without evidence. Two reviewers argued that “evidence-free claim verification” is not a particularly noteworthy formalization because many verification benchmarks can be run without retrieval, and the paper did not sufficiently justify what is fundamentally new beyond separating retrieval-based vs retrieval-free settings; relatedly, they viewed INTRA as an incremental variation on prior internal-state/probing approaches (layerwise probing plus aggregation) with limited conceptual novelty. Reviewers also raised validity concerns that without external grounding, many natural language claims can be ambiguous or context-dependent, and the submission relies heavily on upstream “atomic claim” decomposition/filtering assumptions without fully quantifying their reliability or sensitivity. Finally, one reviewer flagged evaluation completeness issues (single base model, missing runtime/latency comparisons, lack of confidence intervals and stronger verbalized baselines), which the rebuttal partially addressed, but these additions did not fully resolve the core novelty/meaningfulness concerns driving the rejection lean.

**Reviewer Concerns:**

Addressed by the rebuttal: Reviewer 3x56’s main evaluation-completeness concerns were directly addressed: the authors added results on an additional base model beyond LLaMA-3.1-8B, provided runtime/latency measurements to justify the “lightweight” claim, added comparisons to strong verbalized baselines (including GPT-4.1), and reported statistical reliability via bootstrap confidence intervals and error bars (also responding to the Figure-2/claim-position variance request). The rebuttal also partially addressed Reviewer 33RQ’s positioning critique by clarifying that evidence-free verification is intended to be complementary to RAG (not a replacement) and by explaining the reliance on atomic-claim pipelines for handling ambiguity.

Still outstanding: The core concerns driving the rejection lean remain largely unresolved for Reviewers hra2 and 33RQ: (1) novelty/credit for the task framing—i.e., whether “evidence-free claim verification” is a substantive new research problem versus a relatively straightforward reconfiguration of existing benchmarks—and (2) method novelty, as INTRA is still perceived as a modest extension of prior internal-state/probing approaches (layerwise classifiers plus aggregation) without a strong new conceptual insight. In addition, while the rebuttal asserts that ambiguity is handled upstream via decomposition/filtering, the paper still lacks sufficient evidence about the robustness and realism of the atomic-claim assumption (e.g., sensitivity to decomposition quality, prevalence of context-dependent claims), which undercuts claims about broad real-world applicability without retrieval.

**Reviewer Scores:**

Reviewer hra2 (initial: 2 / reject): Likely no change. Their objections are primarily about novelty/credit (task “trivial,” method too close to prior internal-state work) rather than missing experiments; discussion would not fully resolve this, though clarifying robustness across datasets might soften slightly.

Reviewer 33RQ (initial: 2 / reject): Likely small increase (possibly 4). The rebuttal improves positioning (evidence-free as complementary to RAG) and clarifies handling of ambiguity via atomic-claim pipelines, which may move them from strong reject to borderline, but they still question whether the task is a noteworthy standalone contribution.

Reviewer 3x56 (initial: 4 / marginally below): Likely increase to marginal accept (4 → 6). Their concerns were largely experimental completeness (single base model, runtime/latency evidence, confidence intervals, stronger verbalized baselines), and the rebuttal directly adds these, so full discussion would plausibly push them over the acceptance threshold.

---

### Decision · Program_Chairs · 2026-01-26

Reject